# The Challenges and Compatibility of Mobility Management Solutions for Future Networks

Muhammad Mukhtar [1,2], Farizah Yunus [1], Ali Alqahtani [3], Muhammad Arif [4,*], Adrian Brezulianu [5,*] and Oana Geman [6]

1  Faculty of Ocean Engineering and Informatics, Universiti Malaysia Terengganu, Kuala Nerus 21300, Terengganu, Malaysia
2  Department of Computing and Technology, Iqra University Islamabad, Islamabad 44000, Pakistan
3  Department of Networks and Communications Engineering, College of Computer Science and Information Systems, Najran University, Najran 61441, Saudi Arabia
4  Department of Computer Science, Superior University Lahore, Lahore 54000, Pakistan
5  Faculty of Electronics, Telecommunications and Information Technology, "Gheorghe Asachi" Tehnical University, 700050 Iaşi, Romania
6  Department of Computers, Electronics and Automation, Faculty of Electrical Engineering and Computer Science, Stefan cel Mare University, 720229 Suceava, Romania
*  Correspondence: arifmuhammad36@hotmail.com (M.A.); adi.brezulianu@greensoft.com.ro (A.B.)

**Abstract:** Wireless network devices can attain the required level of quality of service (QoS) and maintain connectivity even after detaching from a current point of access. This detachment (mobility) requires various mobility management (MM) mechanisms, which present numerous challenges due to the exponential growth of wireless devices and the demands of users. The network must be heterogeneous and dense to manage such a heightened escalation of network traffic, increased number of devices, and different types of user demands. Such factors will seriously challenge MM solutions, eventually making the networks non-feasible from the dependability, adaptability, extensibility, and power consumption points of view. Therefore, novel perspectives on MM mechanisms are desired for 5G networks and beyond. This paper introduces an innovative discussion of the functional requirements of MM mechanisms for advanced wireless networks. We present comprehensive arguments on whether the prevailing mechanisms perceived by standard bodies attempt to fulfill the stated requirements. We complete this discussion through innovative qualitative evaluation. We assess each of the discussed mechanisms in terms of their capability to fulfill the dependability, adaptability, extensibility, and power consumption benchmarks for upcoming MM schemes. Hereafter, we demonstrate the outcome and the identified gaps/challenges for the planning and deployment of 5G MM frameworks and beyond. Next, we present the capabilities and possible MM solutions to tackle the gaps/difficulties. We complete our discussion by proposing a 6G MM architecture based on defined parameters.

**Keywords:** 5G; QoS; 6G; MM; functional requirements

## 1. Introduction

The meaning of handoff in a telecommunication network is to shift a continuing data or call session from one channel to another; this handoff can be attained with the help of MM mechanisms used in 3G, 4G, LTE, or LTE-A networks. These mechanisms (3G, 4G, or LTE-A MM mechanisms) guarantee the continuity of service and seamless connectivity for the devices when transported from the physical location where they were originally attached. However, in 5G network scenarios, we cannot utilize these inherited MM mechanisms due to the inefficiency, non-compatibility, or even un-usability of these mechanisms. The maturity of 4G and its standards, such as ETSI, IETF, and 3GPP, has provided methods and techniques that can be used for both the flexibility and dependability of wireless networks. Still, these techniques and procedures are not so feasible for (i) heterogeneity-type networks,

(ii) generation of revenue for the industry, and the (iii) softwarization deployment approach. In this respect, researchers [1] also used MM techniques (multi radio access technology (RAT) and software-defined networking (SDN)), but these, too, cannot tackle the MM challenges of 5G and future networks due to the volume of devices, complexity, etc.

Similarly, as noted in the literature [2,3], these MM techniques are only feasible for current architectures and unable to cope with 5G and future networks. Next, the MM techniques of group handover, progressive cell association [4–6], etc., are also not suitable for tackling the issue of complexity and core network signaling in 5G and future networks. Consequently, such unfeasible aspects have forced researchers to initiate a great revolution in MM mechanisms for 5G and future networks. Researchers face issues in molding existing MM to the 5G network. These issues are encountered due to the architectural nature of the 5G network, in which the provision of diverse types of services and satisfying the expectations of users ($120\times$ increase in data rates) are a must compared to the LTE network [6,7]. Moreover, the theme of 5G is to ensure minimum latency with required reliability for delay-sensitive applications, such as emergency services, augmented reality, multimedia applications, real-time applications, etc. [6].

Furthermore, the ITU defined the detailed requirements that 5G networks will have to fulfill [8]. Specifically, the central challenge of the 5G network is to provide services to users at the time of mobility, when the data rate is almost more than 1 Gbps, with heterogeneous applications and mobility profiles and extreme dependency on the network. However, the literature and understanding to date suggest there has been no thorough assessment of the functional requirements, possible solutions, and challenges concerning MM mechanisms. Hence, we discuss the MM mechanisms of 3G, 4G, LTE, LTE-A, etc. and their issues in the 5G network, followed by state-of-the-art MM mechanisms for 5G and future networks. We divide the article into the following sections. Section 2 presents the functional requirements and design principles for 5G and future MM mechanisms. Section 3 lays out the conditions for qualitative analysis and the criteria that must be met for the satisfaction of these conditions. Section 4 presents a qualitative analysis of legacy mechanisms and compares the advantages and drawbacks of 5G and future MM solutions. Next, in Section 5, we analyze current state-of-the-art mechanisms and their usability, challenges, and possible compatibility as MM solutions with 5 G and future networks. Further, in Section 6, we present possible strategies and existing challenges that will help to resolve the issues presented in Sections 5 and 6 with the requirements stated in Section 2. Furthermore, Section 6 also proposes a framework for 6G MM solutions. Lastly, Section 7 concludes with possible future directions.

## 2. Design Principles and Requirements for Upcoming MM Techniques

Upcoming wireless networks will assist in accommodating various tenants (groups of users having common access and privileges) as well as industry verticals on the same network structure [9]. These revolutions, a few of which are being discussed by researchers [6], exemplify a model move from the existing network architecture design. As a result, MM mechanisms will need to be redesigned or re-assessed. We initially present the functional requirements of MM mechanisms for upcoming wireless networks in Table 1, formulated on the basis of features we derived from various network scenarios (current and forthcoming). From Table 1, we can perceive that the MM solutions for 5G and upcoming networks will have to adjust and change in order to cope with future wireless networks competently.

Further, MM solutions will need to be reshaped, so that they may offer adaptability, dependability, extensibility, and reduced power consumption to guarantee unified mobility and required QoS. In one respect, from these requirements, specific criteria will affect the development and design of future MM solutions. Therefore, in the subsequent text, we provide an understanding of these various design benchmarks and their influence on 5G and upcoming MM mechanisms.

**Table 1.** Functional requirements of MM mechanisms.

| Requirement | Ongoing Scenario | The Scenario in 5G and Beyond | MM Requirements |
|---|---|---|---|
| 1 | Complexity is considered based on user requirements in a homogenous network [7] | The complexity is the sum of heterogeneous radio access technology (RAT) scenarios, the dense nature of the network, various QoS requirements of users, and backhaul scenario [10] | 1. Flexibility to accommodate heterogeneity<br>2. Tractable solution (low computational c006Fmplexity)<br>3. Low power utilization for a dense network |
| 2 | Up to 400 km/h mobility support | Mobility support up to 500 km/h [5] | Accommodating various demands of mobility profiles |
| 3 | Services and data are entertained in a core network | Demanded data and services can be entertained at network edges by multi-access edge computing (MEC) [4] | Ensuring QoS service for applications during mobility for service replication [8] and migration |
| 4 | 6 GHz frequency band | Sub 6 GHz, Terahertz communication [11,12], millimeter wave (mmWave) [13] | The level of robustness has a prominent impact on MmWave and (visible light communication (VLC), hence challenging continuous connectivity |
| 5 | Fine granularity of localization and tracking <50 m [14] | Fine granularity of localization and tracking <10 cm [14] | Capability to offer better granularity in a dense or urban environment |
| 6 | MM protocols are standardized for 2G to 4G networks and devices | These are different from 2G to 4G | Support of backward Compatibility |
| 7 | Radio tower-based (static) network | Relay stations and BS may be used on drones in 5G and future networks [15,16] | Mobility provision for BS and UE |
| 8 | Connectivity in the range of radio access network (RAN). | Connectivity between RANs | Provisions for efficient radio access technology (RAT) selection process as well as multi-RAT MM support |
| 9 | The network is driven by a vendor [4] | The network has become softwarized [4] | Capability to use MEC, software-defined network (SDN), near-field communication (NFC), etc. |
| 10 | The density of user equipment is nearly 100,000 users per km$^2$ | The density of user equipment is nearly 1,000,000 users per km$^2$ | Support of mobility for an increasing number of users |
| 11 | Provision of services for mobile broadband applications | There is support for applications with different QoS requirements such as ultra-reliable low latency communications (URLLC), massive machine type communications (mMTC), and enhanced mobile broadband (eMBB) applications | Provision of mobility support according to the context |
| 12 | The complexity of the network is handled and operated by a human being | Data traffic is booming, with more complex networks and devices [17–19] | Low energy consumption in such a complex network |
| 13 | The traditional optimization for spectrum range | Complex optimization for new spectrum range [20,21] | Low energy consumption by using a new spectrum range |

We conclude from Table 1 that using only one MM solution will be inefficient to tackle all prevailing and future circumstances. Hence, these solutions face the challenges of attaining careful IP packet dispatching, RAT and BS selection, session control, and path optimization. Further, some other applications such as simulated reality, vehicle-to-vehicle (V2V), vehicle-to-infrastructure (V2I), improved reality, etc., will require high bandwidth, dependability, and minimum delay requirements [22]. These requirements will also become a challenge for MM mechanisms. Additionally, RAN technologies are

also required to transform future networks [2]. This will also have adverse effects on the QoS parameters at the time of roaming, which will also be elaborated on in later sections. Likewise, we also face another concern (challenge) for the MM mechanisms, especially when the base station is moved, such as in drones [23] or a squad of vehicles moving together or joining and disjoining vehicles or extremely fast user movement (200 to 600 km/h), etc. Furthermore, the purpose of the latest MM techniques is to embed software (SDN) for controlling the mobility between the RATs [24]. Still, these embedded techniques have not been particularized in some other encountered aspects, such as the burden of the number of messages exchanged, time convolution, etc. Moreover, advanced cell association and group handover-based MM mechanisms are discussed to cope with heterogeneity in profiles and mobility patterns in the 5G network. These mechanisms lack in terms of tackling the complexity, network burden, etc., that will be faced by 5G and future MM mechanisms.

Consequently, we have reached a new dimension, wherein MM solutions are required to be distributed, flexible, and able to maintain various use cases concurrently and keep account of the numerous other significant changes in 5G networks with dependability. Consider that decentralization will allow MM mechanisms to facilitate increases in attached devices with different mobility profiles (e.g., stationary IoT devices and users in high-speed vehicles). On the other hand, flexibility will permit them to adapt to the user, environment, and network situation (e.g., network potential, QoS, user mobility profile, type of flow, etc.). In this context, we also need to consider the following design concerns regarding MM mechanisms for 5G and future networks [25,26]. These are as follows: (i) signaling (control plane) concern [27], (ii) backhaul concerns [28], (iii) resources computation concern [15,29], (iv) concerns over physical layer (v) provision of service granularity [16,30] and (vi) perspectives of parameters [24]. A comprehensive refurbishment of MM mechanisms for upcoming wireless networks will require quite a long period and effort to attain the optimum solution. Therefore, in the subsequent sections, we carry out an innovative qualitative analysis of several legacy techniques, current mechanisms, and calibration efforts and assess their suitability as enablers for MM in 5G and future networks.

## 3. Qualitative Benchmark for Analysis

The current MM mechanisms are suitable for the 2nd, 3rd, and 4th generation, but cannot perform well for 5G and later network scenarios. Hence, there is a need to explore whether these mechanisms can be applied partially or as a whole to the stack of 5G and future networks. So, in this section, we analyze these MM mechanisms based on their usage of energy, adaptability, dependability, and network extensibility. As a portion of this analysis, initially, we present a thorough explanation of these criteria, as follows.

**(i) Adaptability** is a way of qualitative analysis in which we measure the adaptability aspects of the MM mechanisms for the network. **(ii) Dependability** will decide whether the MM mechanisms will be capable of confirming uninterrupted services in any network. **(iii) Extensibility** feature lets one define if the MM mechanism can offer the services to the increased number of devices with a parallel increase in demanded QoS with a heterogeneity mobility profile. **(iv) Energy efficiency** concerns whether portable devices are limited in storage and energy due to their petty design. The energy efficiency will determine whether the MM mechanisms will be capable of ensuring uninterrupted services in any network with prevailing power resources [31].

We arranged the above benchmarks (i, ii, iii, iv) for analysis into a list of parameters for each criterion and present them in Table 2. We present the acquiescence with each prescribed parameter in Table 2 for extensibility, adaptability, dependability, and energy efficiency. Now, we elaborate upon the parameter requirement relationships that have been shown in Table 1 in order to improve the completeness of the assessment criteria.

**Table 2.** Major parameters for the dependability, adaptability, extensibility, and power consumption of a MM mechanisms/standards.

| Sno | Dependability | Requirement Consideration | Sno | Adaptability | Requirement Consideration | Sno | Extensibility | Requirement Consideration | Sno | Power Consumption | Requirement Consideration |
|---|---|---|---|---|---|---|---|---|---|---|---|
| DL1 | Consideration of congestion | 2 | AL1 | Multiple parameters for handoff decision | 1, 9 | EL1 | Decentralization | 4 | PL1 | Signaling load due to frequent handoffs | 12 |
| DL2 | Decentralization | 3, 4 | AL2 | Handoff facility provision at all levels | 4, 9 | EL2 | Processing load with increased numbers of users | 3, 9 | PL2 | Increased signaling load due to increased users | 1, 12 |
| DL3 | Re-routing at CN, fast path | 5, 10 | AL3 | Context awareness | 2, 9, 10 | EL3 | Signal load tackling with an increased number of users. | 3, 9 | PL3 | Increased link failure | 1, 12 |
| DL4 | Seamless handover | 1, 7, 8 | AL4 | All-time connectivity by multiple APs | 1, 9 | EL4 | Ease of implementation and integration. | 6 | PL4 | Congestion | 1, 12 |
| DL5 | Repetition in the number of connections and flows | 7 | AL5 | Service granularity per flow, service and user. | 9, 11 | EL5 | Connection management at the time of increased users | 3, 9 | PL5 | Processing load due to an increased number of users | 1, 12, 13 |

### 3.1. Criteria for Requirement Mapping for Dependability

The criterion for meeting the DL1 parameter will assist in satisfying the 11th requirement of Table 1 in Section 2. Similarly, the DL3 parameter contains significant associations toward satisfying requirements 3 and 7, as mentioned in Table 1. Likewise, the DL3 parameter also ensures that data path changes due to service replication do not lead to delays. Next, the DL4 parameter will help to achieve requirements 4, 5, and 8 as mentioned in Table 1. Here, the capability to offer seamless handover support in providing mobility between multiple RATs is mentioned in requirement 8 of Table 1. Likewise, improving localization proficiencies to complete the same in a dense and urban environment is mentioned in requirement 5 of Table 1, and lastly, correlating multi-connectivity and thus attaining the dependability is mentioned in requirement 4 of Table 1. Lastly, the criterion for providing redundant flows and connections may be met by satisfying the parameter DL5, which can fulfill requirement 4 mentioned in Table 1.

### 3.2. Criteria for Requirement Mapping for Adaptability

By satisfying parameter AL1, the MM mechanism (understudy) fulfills requirements 1st, 8th, and 12th of Table 1. Similarly, when the AL2 parameter is considered, it assists in achieving the 1st and 9th requirements of Table 1 [32]. Further, when the condition of the AL3 parameter is fulfilled, then it will satisfy requirements 1, 3, and 11 of Table 1. Furthermore, parameter AL4 will assist in fulfilling requirements 1 and 8 of Table 1. Lastly, the requirements mapping criteria for the MM mechanism are considered to fulfill the adaptability parameter AL5, congruently assisting in achieving the requirements 1 and 2 mentioned in Table 1.

### 3.3. Criteria for Requirement Mapping for Extensibility

The criterion for requirement EL1 for extensibility is met, which allows the handover provision at various levels of the network, utilization of MEC, NFV, and SDN platform required for the proficient implementation of requirement 9 mentioned in Table 1. At the time of considering the MM mechanism, the criteria for the requirement mapping for fulfilling the extensibility parameters EL2, EL3, and EL5 congruently are met in order to assist in achieving requirements 1 and 10 of mentioned in Table 1. Finally, when the criterion for parameter EL4 is satisfied, requirement 6 of Table 1 is achieved.

### 3.4. Criteria for Requirement Mapping for Energy Efficiency

The following criteria (requirements 1 and 12 of Table 1) are used for requirement mapping to fulfill the energy utilization parameters PL1, PL2, and PL3 congruently because adding numerous power-sensitive activities to the edge server will increase the battery life by 70 percent, as stated in requirement 1 of Table 1. Similarly, various offloading algorithms and applications, such as a genetic procedure-based and multi-objective workflow-based computation offloading framework that takes care of task completion constraints and energy usage, will try to decrease energy usage. Similarly, identity is another multimedia-based framework that can handle 50 times more computational load than other applications by manipulating the task of offloading, as mentioned in requirement 12 of Table 1. Next, meeting the criteria for parameter PL4 of the power sensitivity provides handover options at different levels of the network and usage of SDN, MEC, and NFV platforms that are needed for the efficient application of requirement 12 mentioned in Table 1. Likewise, dynamic multipath optimization (DMPO) reduces the offloading path after user movement. It forecasts the next position and user data size for the short term but does not perform well when the distance among the stations (base stations) is irregular.

Similarly, enterprise mobility management (EMM) segregates information into two distinct categories. The first category covers the channel state information, and the second covers the geographical position, number of base stations, and user task size. The energy usage at the time of handoff is decreased based on these mentioned categories, but it performs worse when multi-user systems are involved, as mentioned in 12 of Table 1.

Finally, when parameter PL5 is considered, it assists in achieving requirement 13 of Table 1. The rationale is that heuristics-based methods are most appropriate for high-speed vehicles.

Additionally, according to the above-mentioned mapping, it can be concluded that the selected criteria for our qualitative analysis seemed to be more comprehensive than earlier work in this domain. Further, we considered the key performance indicator (KPI) [33] of the 5G network. These indicators can be fulfilled with the help of dependability and by the use of efficient energy utilization metrics during mobility scenarios. Hence, more than 90 percent dependability and 25 percent battery saving are ensured. Similarly, with the help of dependability and adaptability metrics, the latency can be reduced for virtual reality, broad-based applications, and connected cars up to 10 ms and 5 ms, respectively. The dependability metric also helps to provide reliable link selection and traffic conditions on the network. In addition, there are approximately 0.75 million devices/km$^2$ with multiple applications and mobility profiles that can ensure the criteria of extensibility and battery-saving. Now, in the following sections, we can further evaluate and judge the criteria of the aforementioned qualitative analysis.

## 4. Suitability of Legacy Techniques and Standards for 5G and Upcoming Networks

We assess the most widely (commonly) used legacy techniques and standards based on the parameters (adaptability, dependability, extensibility, and energy saving) as mentioned in Table 2. The aim of this analysis is not to compare the legacy techniques with themselves but to evaluate them for their applicability in 5G and future networks, in the light of mentioned parameters in Table 2. The role of each technique and standard is provided as follows.

### 4.1. Proxy Mobile IPv6 (PMIPV6)

The structure of proxy mobile IPV6 is such that it does not need any involvement from a mobile node in the handover process [30,34,35]. In light of the discussion in Section 3, we identify the advantages and drawbacks of PMIPV6 techniques concerning their usage in 5G and future networks. These are as follows:

**Advantages**

-   The decentralization can be achieved by supporting the DMM approach of PMIPV6 [35].
-   Similarly, by supporting the cluster-based technique in PMIPV6, the dependability, energy-saving, and extensibility can be enhanced [25].
-   Likewise, seamless mobility is also possible with the support of [25,29].
-   Furthermore, the stated approach (PMIPV6) is also adopted by 3GPP for LTE; hence, the pertinent implementation capability will increase the ease of usage for the adaptation of the future network.

**Drawbacks**

-   An apparent behavior of PMIPv6 concerning the parameters for adaptability and battery-saving criteria is omitted [29,35].
-   PMIPv6 also lacks the capabilities of adaptability and dependability in its original specification due to the creation of single point of failure (SPOF) molded by the LMA in its design [25].

In light of the stated advantages and drawbacks of PMIPv6, it is inferred that the seamless handover parameter DL4 (Table 2) in dependability criteria is fulfilled by the original implementation and deployment of PMIPV6. Similarly, with the help of distributed and cluster-based approaches in PMIPV6, the parameters EL2 (Table 2) and DL5 (Table 2) of extensibility and dependability are fulfilled, respectively. Finally, as we also know that PMIPV6 has been applied and explored in the LTE networks, it satisfies the parameter EL4 (Table 2) of extensibility that is based on ease of implementation.

*4.2. Multipath TCP (MPTCP) and Stream Control Transmission Protocol (SCTP)*

MPTCP is a continuing struggle of the Internet Engineering Task Force's (IETF) working group, whose objective is to permit a TCP connection to utilize multiple paths in order to capitalize on resource usage and raise redundancy [36–38]. SCTP provides multipath congestion awareness and redundancy, like in MPTCP [39]. Furthermore, SCTP is also a prospective enabler for the latest MM mechanisms [40]. In order to judge the compatibility of MPTCP and SCTP for 5G and future networks, we list their advantages and drawbacks in the following section [41].

**Advantages** (MPTCP)

- Multiple sub-flows can be handled independently [35,42]
- We have a specific congestion control mechanism for each flo
- There is flexibility against connection failure due to the division of flows [23,42]

**Drawbacks** (MPTCP)

- Lack of optimized middleboxes for supporting the MPTCP [23]
- Proxies are needed to take full advantage of MPTCP architecture [43,44].

**Advantages** (SCTP)

- With the support of a multi-homing feature, the network-level fault tolerance can be achieved [24].
- The congestion awareness is embedded to avoid the SCTP suite's congestion [24].
- This approach (SCTP) permits the execution of multiple connections at the transport layer [24]. Hence, connection failures can be tackled by shifting the control to other available flows.

**Drawbacks** (SCTP)

- We need to update the existing protocols of all SCTP-enabled communicating devices with the SCTP protocol [24].

As a result, from the above findings on MPTCP and SCTP, it is revealed that IETF MPTCP-SCTP fulfills parameter AL5 (Table 2) by offering the service granularity per flow, per service, and per user. Similarly, MPTCP-SCTP approaches meet the criteria DL5 of Table 2 (allowing the single user several flows over the network) and DL1 of Table 2 (consideration of congestion provisioning awareness as part of MM characteristics) for the dependability criterion.

*4.3. IEEE 802.21*

This standard provides a conjoint policy where the upper layer can interact with the lower layers [28,45]. Next, this standard offers media-independent handover (MIH) services, including events, commands, and information services [46]. By using the stated benefits, this protocol facilitates the upper layers in the protocol stack to acquire information that is obtainable on MAC and link layers for guaranteeing continuous connectivity among domains and, therefore, improves the quality of experience and quality of service [27–29]. We identify the following advantages and drawbacks in the light of defined parameters in Table 2.

**Advantages**

- There is an option for user equipment (UE) to attach to more than one access point [28,47].
- The UE can also switch between various RATs [47–49] with seamless handoff capability.

**Drawbacks**

- We need to update the existing protocol stack so that communicating devices may be able to use the features of the IEEE 802.21 standard [26,49].

Hence, in light of the stated discussion regarding IEEE802.21, it can be inferred that the IEEE802.21 standard fulfills AL4 (Table 2) requirements for the adaptability (privilege

of connection with more than one access point) point of view. Similarly, the DL4 (Table 2) parameter of dependability (privilege of seamless handover among various access points) is also fulfilled by this standard.

### 4.4. Received Signal Strength (RSS) Technique for AP Selection

Received signal strength (RSS) selects the base station by comparing the sensed BS link features, such as reference signal received quality (RSRQ), RSRP, or RSSI level [25,50,51]. The said method is simple and easy to implement and does not require higher signaling or processing overhead. However, some concerns may afflict such a method, as BSS with good RSS value may be overburdened, while the rest of the stations may not be adequately utilized. To review the RSS-based AP selection technique, we present the advantages, drawbacks, and discussion in light of the defined parameters in Table 2.

**Advantages**

-    The RSS approach is simple and straightforward because a small number of signals are exchanged and low processing is required.
-    Due to its simplicity, its implementation is also straightforward, as it is already being used by 3GPP [30,52].

**Drawbacks**

-    This approach is unfeasible in dense environment (most populated) [51], especially during frequent handoffs.
-    The approach is one-dimensional and unreliable, owing to the uncertainty regarding the context of a user, network load, network heterogeneity, etc. [30,51,52].

In light of the discussion, the RSSI-based technique is simple in its implementation and deployment point of view. Hence, it fulfills the parameters EL1, EL3, and EL5 of Table 2 concerning extensibility. Similarly, it also meets the parameters mentioned in PL2 and PL5 of Table 2, an energy-saving point of view. Next, the RSSI-based technique can also guarantee the movement between multiple RATs, fulfilling the parameter DL3 of Table 2 for dependability.

### 4.5. LTE MM Mechanisms (3GPP)

The LTE MM mechanisms (concerning functionality) can further be segregated into the following subcategories: (i) offload traffic in 3GPP, (ii) LTE-Wi-Fi accumulation, (iii) twofold connectivity, and (iv) LTE handoff. The discussion and applicability of these MM mechanisms regarding the concerned parameters are as follows.

i.      Offload traffic flow in 3GPP

The aim of traffic offloading is to support the operators so that the traffic burden may be lessened on the core network [53,54].

**Advantages of Offloading traffic flow in 3GPP**

-    By this technique, the processing burden is managed on the nodes gracefully.

**Drawbacks of offloading traffic flow in 3GPP**

-    In the scenario of communication with a non-3GPP network, it is a bit difficult to deploy IFOM for such coordination with a non-3GPP networor.
-    An extra gateway is required by local IP access (LIPA) for its implementation.
-    LIPA cannot support service continuity at the time of handoff.
-    Radio resources congestion cannot be moderated with the help of SIPTO [36].

ii.     LTE Wi-Fi Accumulation and Twofold Connectivity

This user equipment (UE) can receive packets from LTE and Wifi interfaces concurrently [29,55].

**Advantages:**

- This approach has the option to use multiple physical paths, as well as the characteristics to tolerate the faults [56].
- This approach also has the option for making connections with RAT of non3GPPP as well as with 3GPP [57].

**Drawbacks:**

- LTE WI aggregation (LWA) is only suitable for downlink.

iii. LTE Handoff

According to the context of the draft, we attained the following advantages, drawbacks, and discussion from the LTE handoff mechanism [58].

**Advantages**

- The handoff provision is available at the core and access network due to S1 and LTE-X2 methods.
- The CN signaling can be evaded with the help of LTE-X2 handoff metho.
- The capability of LTE-X2 allows it to decide handoff at the access network level; therefore, it provides fast handoff and reduces the processing load on the corresponding nodes.

**Drawbacks**

- The SPOFs are introduced along with the increase in processing load on the CN.

Owing to the mentioned discussion, the LTE MM mechanisms provide improved support concerning the extensibility criterion for 5G and future networks. They also fulfill parameters EL1 to EL4 by the features of decentralization, simplicity of incorporation, traffic offloading features, and multi-level handoff methods, i.e., S1, and X2 handover [59]. Similarly, it is evident that duplication in data routes (through DC and LWA), decentralization (through X2 and traffic offloading), and seamless handover (through X2 and S1 handover) also fulfill the criteria of dependability in parameters DL2, DL3, and DL4, respectively, as mentioned in Table 2 of Section 3. Next, for the adaptability criteria, LTE MM mechanisms provide an option of joining multiple access points simultaneously, hence satisfying parameters DL3 and FL4 for adaptability as mentioned in Table 2 of Section 3. Lastly, it also provides improved support concerning the power-saving criterion for 5G and future MM mechanisms by fulfilling parameters PL1 and PL3 mentioned in Table 2 of Section 3.

*4.6. LTE Multi-Connectivity Solutions (Non-3GPP)*

There are some standards and methods (separate from 3GPP) that use the notions ITU-vertical multihoming (ITU-VMH) and the coordinated multipoint (CoMP) [60–62]. In light of the discussion presented in Table 2 of Section 3, we identified the following advantages and drawbacks of multi-homing and COMP approaches concerning their usage in 5G and future networks. These are as follows.

**Advantages of Multi-Homing Support**

- There is the possibility of channel-based granularity of service.
- At any instance of time, there is the possibility of connection to more than one access point [63].
- Redundancy can also be attained [63]

**Drawbacks of Multi-Homing Support**

- The protocol needs to be updated [63].

**Advantages of COMP**

- At any instance of time, there is the possibility of connection to more than one access point [60,64].
- There is the possibility of channel-based granularity of service.

- Redundancy can also be attained [64].

**Drawbacks of COMP**

- To sustain the low delay and high capacity of the backhaul network, the need to conserve QoS with the support of COMP is a real challenge.
- Maintaining consolidated processing is another key challenge [60].

Owing to the mentioned discussion, the non-3GPP multihoming and COMP mechanisms provide improved support for adaptability criteria, such as AL5 (having the option of granularity on a per-channel basis for MM mechanisms) and AL4 (permitting the probability of connecting to multiple access points). Similarly, the criteria for 5G and future MM mechanisms for dependability are fulfilled with the help of parameters DL4 (permitting unbroken and seamless connection at the time of handover) and DL5 (permitting the option of redundant physical connections).

Eventually, we will conclude in Section 4 by summarizing the findings in Table 3, wherein we have indicated all of the explored mechanisms in the light of dependability, adaptability, extensibility, and power-saving criteria. Furthermore, we have also included the references in Table 3 for supporting the analysis and arguments. The review, discussion, and Table 3 show that none of the legacy mechanisms could satisfy and cope with the requirements of 5G and future network in light of the considered parameters. However, 3GPP-based LTE MM mechanisms facilitate better on the basis and provision of 5G and future MM mechanisms, provided that they jointly fulfill the maximum parameters amongst other considered techniques. Furthermore, with the help of qualitative analysis, we have formed conclusions regarding the worth and challenges of legacy mechanisms for 5G and future MM networks in the light of parameters mentioned in Table 2 of Section 3. Specifically, we concluded that legacy MM techniques are insufficient in terms of the stated worth and challenges. Hence, we investigate novel state-of-the-art MM solutions for 5G and future networks in the subsequent section.

**Table 3.** Compliance with dependability, adaptability, extensibility, and power saving.

| Considered Approaches | Dependability | | | | | | Adaptability | | | | | | Extensibility | | | | | | Power Saving | | | | | |
|---|---|---|---|---|---|---|---|---|---|---|---|---|---|---|---|---|---|---|---|---|---|---|---|---|
| | DL1 | DL2 | DL3 | DL4 | DL5 | RF | AL1 | AL2 | AL3 | AL4 | AL5 | RF | EL1 | EL2 | EL3 | EL4 | EL5 | RF | PL1 | PL2 | PL3 | PL4 | PL5 | RF |
| 1. Handover (RSS based) | 0 | 0 | 0 | 1 | 0 | [30,50,52] | 0 | 0 | 0 | 0 | 0 | [28,30,51,52] | 0 | 1 | 1 | 1 | 0 | [30,50,52] | 0 | 1 | 0 | 0 | 0 | [17] |
| 2. Multi-connectivity methods (non 3GPP based) | 0 | 0 | 0 | 1 | 1 | [63,64] | 0 | 0 | 0 | 1 | 1 | [63,64] | 0 | 0 | 0 | 0 | 0 | [3,60] | 0 | 1 | 0 | 0 | 1 | [18,19] |
| 3. LTE MM Mech | 0 | 1 | 0 | 1 | 1 | | 0 | 1 | 1 | 0 | 0 | [65] | 1 | 1 | 1 | 1 | 0 | | 1 | 0 | 0 | 0 | 1 | [18,19] |
| 4. IETF PMIPV6 | 0 | 1 | 1 | 0 | 0 | [29,35,51] | 0 | 0 | 0 | 0 | 0 | [30,35] | 1 | 0 | 0 | 1 | 0 | [3,29,66] | 0 | 0 | 0 | 1 | 0 | [18] |
| 5. IEEE 802.21 | 0 | 0 | 0 | 1 | 0 | [51,67] | 0 | 0 | 0 | 1 | 0 | [28,47,48] | 0 | 0 | 0 | 0 | 0 | [26,49] | 0 | 0 | 1 | 0 | 0 | [19] |
| 6. IETF MPTCP/SCTP | 1 | 0 | 0 | 0 | 1 | [67,68] | 0 | 0 | 0 | 0 | 1 | [42,69] | 0 | 0 | 0 | 0 | 0 | [23,43] | 0 | 0 | 0 | 0 | 1 | [18] |

1 means the condition of the given parameter is satisfied, 0 Means the condition of the given parameter is not satisfied, RF means reference.

### 5. Recent MM Techniques and 5G Architecture

The following text highlights the MM mechanisms tailored with 5G architecture. We segregate the section into the following areas: (i) Design aspects of 5G-related MM mechanisms, (ii) Research contribution (non-3GPP) for MM strategy in 5 G network, and (iii) MM mechanisms and 6G network.

*5.1. Design Aspects of 5G Related MM Mechanisms*

The literature [33,70,71] presents the development and design aspects of MM mechanisms in the 5G network. These mechanisms (on the basis of functionality) can be categorized as follows: (i) monitoring of user mobility [33], (ii) provision of multi-homing [33], (iii) novel session controlling methods, (iv) coverage of user plan [33], (v) twofold association and computation at edges [33], (vi) slicing in networks [22], (vii) congestion control and load balancing technique [33], (viii) new MM module (inter and intra next-generation handover) [33,70], and (ix) device to device (D2D) mobility provision [72]. Furthermore, the aforementioned techniques can be enhanced by introducing the global view of network scenarios and softwarized solutions for 5G and future MM methods. Subsequently, we discuss these innovative MM mechanisms for 3GPP in the following text. Now, in light of the discussion in Table 2 of Section 3, we identify the advantages and drawbacks of 3GPP MM techniques in 5 G networks. These are as follows.

**Advantages**

- IPV6 multi-homing support [27]
- Facility of service stability modes for various sessions [27]
- Facilitation of event notifications, resource negotiation mechanisms, and monitoring of UE mobility at target RAT [27]
- Provision of per-PDU session granularity, traffic direction, malleable session control, and roaming option [27]
- Provision of handover for inter-RAT and intra-RAT [27]
- Sound and well-defined 5GC inter-working interface, i.e., N26 [27] and EPC
- D2D level mobility provision [55,71]
- Provision for multi-RAT DC [27]
- Provision for edge computing [27]
- The network slicing information movement provision at the time of intra/inter-RAT mobility [27]
- Network slicing provision for supporting on-demand MM
- Potential to offer context cognizance via the slicing of the network [11]
- Potential for handling core network burden by offering re-balancing philosophies and load balancing on the AMF [27]

**Drawbacks**

- The harmonization among D2D peers for sanctioning the well-organized MM techniques has not been explored.
- Signaling at the time of handoff is not so optimal in the correspondence node.
- Still, the fundamental approach for discovering any access point is based on received signal strength (RSS) [73].
- An integrated structure for cross-layer approaches, such as MR-DC (physical and MAC layer), MPTCP-SCTP (transport layer), and IPv6 multi-homing (network layer) working collectively, has not been explored.
- Due to the single occurrence of user plan functionality (UPF) in IPV6 multi-homing, there is still a chance of SPoF [27].

According to the identified advantages and drawbacks of 3GPP 5G MM mechanisms, we can justify the completeness of the parameters (dependability, adaptability, extensibility, and power consumption) in the following text along with the provision in Table 4.

**Table 4.** Compliance with dependability, adaptability, extensibility, and power saving for 5G MM solutions.

| Research Contribution | Solutions | Dependability | | | | | | Adaptability | | | | | | Extensibility | | | | | | Power Saving | | | | | |
|---|---|---|---|---|---|---|---|---|---|---|---|---|---|---|---|---|---|---|---|---|---|---|---|---|---|
| | | DL1 | DL2 | DL3 | DL4 | DL5 | RF | AL1 | AL2 | AL3 | AL4 | AL5 | RF | EL1 | EL2 | EL3 | EL4 | EL5 | RF | PL1 | PL2 | PL3 | PL4 | PL5 | RF |
| 3GPP based | 1. 3GPP-5G MM Mech | 1 | 1 | 0 | 1 | 1 | [27,55,71] | 0 | 1 | 1 | 1 | 1 | [27,55,71,73] | 1 | 0 | 0 | 1 | 1 | [27,55,73] | 1 | 1 | 0 | 0 | 1 | [17] |
| Core network centered | 2. SDN | 1 | 1 | 1 | 1 | 0 | [72] | 1 | 1 | 0 | 0 | 1 | [72,74] | 1 | 1 | 1 | 0 | 1 | | 0 | 1 | 0 | 0 | 1 | [17,19,21] |
| | 3. DMM | 0 | 1 | 0 | 1 | 0 | [10,44,75,76] | 0 | 0 | 0 | 0 | 1 | [10,75] | 1 | 1 | 1 | 0 | 1 | [10,76] | 1 | 0 | 0 | 1 | 0 | [19,21] |
| | 4. Edge cloud | 0 | 1 | 0 | 1 | 0 | [53,77] | 1 | 1 | 1 | 0 | 1 | [53,77,78] | 1 | 1 | 1 | 0 | 1 | [53,77] | 0 | 1 | 0 | 0 | 0 | [21,79] |
| Access network centered | 5. Phantom cell | 0 | 0 | 0 | 1 | 1 | | 0 | 1 | 0 | 1 | 1 | [6] | 0 | 1 | 1 | 1 | 1 | [27] | 1 | 1 | 1 | 0 | 0 | [21,79] |
| | 6. RANAA service approach | 1 | 0 | 0 | 1 | 0 | [80,81] | 1 | 1 | 0 | 1 | 1 | [80,82,83] | 0 | 1 | 1 | 0 | 1 | [80,82,83] | 0 | 0 | 0 | 1 | 1 | [2,17] |
| | 7.cross Layer | 1 | 0 | 0 | 1 | 0 | [63,84–86] | 1 | 0 | 0 | 1 | 0 | [63,84,85] | 0 | 0 | 0 | 0 | 0 | [84,85,87] | 0 | 0 | 0 | 1 | 0 | [17,21,79] |
| | 8. Smart RAT | 0 | 0 | 0 | 1 | 0 | [27,88–90] | 0 | 0 | 1 | 1 | 1 | [88,90,91] | 0 | 0 | 0 | 1 | 0 | [27,88–90] | 1 | 0 | 0 | 1 | 0 | [1,21] |
| Extreme edge network centered | 9. Device to Device | 0 | 0 | 0 | 1 | 0 | [92–94] | 0 | 1 | 0 | 0 | 0 | [92–94] | 1 | 0 | 0 | 0 | 0 | [92–94] | 0 | 0 | 0 | 0 | 0 | [17] |

1 means condition of the given parameter is satisfied, 0 means the condition of the given parameter is not satisfied, RF means references.

(i)   Dependability parameter considerations

The following parameters of dependability fulfill the criteria for MM mechanisms for 5G and future networks based on our understanding and review of the literature. These are:

(a)   DL1: This parameter can fulfill the criteria by providing a congestion awareness feature in NAS.

(b)   DL2: This parameter can fulfill the criteria by introducing decentralization in MM mechanisms.

(c)   DL4: The criteria of ensuring the seamless handoff capability with the help of MR-DC and handoff mechanisms.

(d)   DL5: This parameter fulfills the dependability criteria by providing multiple flows and connections.

(ii)   Adaptability Parameter Considerations

The following parameters of adaptability fulfill the criteria for MM mechanisms for 5G and future networks in accordance with our understanding and review of the literature. These are:

(a)   AL2: This parameter can fulfill the adaptability criteria by providing handover support at the core, access, and edge network through Xn handover, N2 handover, and 3GPP ProSec, accordingly.

(b)   AL3: This parameter can fulfill the criteria of adaptability by considering the context of the occupant with the help of network slicing.

(c)   AL4: This parameter can fulfill adaptability criteria by making connections to various access points (APs) through IPv6 multi-homing provision and MR-DC.

(d)   AL5: This parameter can fulfill the criteria of adaptability by providing granularity of service offered to maintain per-user mobility level, the capability to support on-demand MM, and per-user PDU sessions with the help of network slicing provision.

(iii)   Extensibility Parameter Considerations

The following parameters of extensibility fulfill the criteria for the MM mechanism for 5G and future networks. These are:

(a)   EL2 and EL3: These parameters can fulfill the criteria of extensibility by the provision of both control plane and data plan-related strategies at the core, access, and edge network concerning MM's point of view.

(b)   EL4: This parameter can fulfill the criteria of extensibility owing to the standardization, hence, no hindrances occur in the deployment, implementation, and integration of RATs.

(c)   EL5: This parameter can fulfill the criteria of extensibility owing to the granularity in service per mobility level.

(iv)   Power Saving Parameter Considerations

The following parameters of power-saving (battery optimization) fulfill the criteria for MM mechanisms for 5G and future networks. These are:

(a)   PL1: This parameter can fulfill the criteria of power-saving owing to the addition of other micros that open for the options of offloading the traffic among the nodes, and every single node can move to the idle state for an extended period.

(b)   PL2: This parameter can fulfill the criteria of power-saving owing to the permission for both more profound and more extended periods of sleep when there is minimal or no continuity in data transmissions.

(c)   PL5: This parameter can fulfill the power-saving criteria by placing the base station into a sleep state with no traffic. Hence, switching off hardware components will ultimately save energy.

*5.2. Research Contribution (Non-3GPP): MM Strategies in 5G Network*

The key dimension of current industrial and academic research is to offer such techniques that can be utilized for tackling user mobility issues, continuity of service, and on-demand-based service provision [95,96]. These techniques can be broadly classified into the following levels: (i) network contribution at the core, (ii) access, and (iii) extreme edge network level.

5.2.1. Network Contribution at Core Level

These (core network) solutions can further be classified as:
(i) Distributed MM (DMM) or edge cloud-based [33,35,53,70,76,97–99]
(ii) Software-defined network (SDN) based solutions [74,75].

In light of the discussion in Table 2 of Section 3, we identify the advantages, drawbacks, and considered aspects of state-of-the-art (non-3GPP) MM mechanisms in the 5G network. These are as follows:

**Advantages of DMM-based mechanisms**

- This solution is helpful at the time of mobility for the corresponding node (CN) to select the efficient data path [10,75,89,100].
- MM anchors can be decentralized by using this mechanism [75,76,89,101].

**Drawbacks of DMM-based mechanisms**

- The coordination and integration with existing devices is a challenging task [10].
- The entire (complete) decentralized mechanism creates a lot of message exchange. Hence, the network is burdened [75].

**Advantages of SDN-based mechanism**

- It provides the overall vision of the network [9].
- Decentralization is possible due to hierarchal solutions [72].
- It allows the facility to design the MM mechanisms according to the context.

**Drawbacks of SDN-based mechanism**

- Wide-range of signals are exchanged at CN [72].
- There are chances of SPOF in DP.

(A)  Considerations for DMM-based Solutions

In dependability aspects, DMM-based solutions fulfill parameters DL2 due to its decentralized nature and DL4 owing to its seamless handoff characteristics. The DMM-based solutions fulfill only parameter AL5 (it offers the granularity of service by preventing any mobility anchor) in the adaptability aspect. Similarly, concerning the extensibility point of view, the DMM-based solutions work like SDN-based solutions by fulfilling the parameters EL1, EL2, EL3, and EL5 for the same intention. Lastly, the DMM-based solutions fulfill parameters PL1 (by putting the base station into a sleep state when there is no traffic) and PL4 (by adding other micros, which will aid in offloading the traffic from the current base station) with respect to the power saving aspect.

(B)  Considerations for Edge Cloud-based MM Solution

In the edge cloud-based MM solutions, parameters DL4 (permitting smooth mobility carried on by fast access to process the capabilities upon movement to the targeted access point) and DL2 (enabling decentralization of MM-based services) fulfill the dependability condition. Next, for the adaptability condition, parameters AL5 (owing to the capability that offers the services based on application profiles and mobility), AL2 (by permitting MM mechanisms at the edge network level in addition to the access and core network-based solutions), AL1 (by providing processing capabilities for user association /AP selection services) and AL3 (by permitting context awareness in data caching concerning user movement) are satisfied. Furthermore, for the extensibility condition, the edge cloud solutions fulfill parameters EL1, EL2, EL3, and EL5. The motivation is that they permit

decentralization, allowing the improved capability to cope with control messages and connections due to the growing number of devices. Finally, the PL2 parameter can fulfill the criteria of the power-saving condition owing to the permission of both deeper and longer periods of sleep when there is minimal or no continuity in data transmissions, which has a substantial effect on the whole network from an energy consumption point of view.

(C)   Considerations for SDN-based MM Solutions

The SDN-based solutions fulfill parameters DL4 (permitting smooth handoff), DL2 (by offering the decentralized mechanism), DL3 (the capability to re-write the code of paths in CN by the organization of guidelines), and DL1 (the ability to use network information for traffic routing with the CN) for satisfying the criteria of dependability. In meeting the adaptability condition, the SDN-based approach fulfills the parameters AL5, AL2 and AL1. Next, in respect of meeting extensibility conditions, SDN-based solutions fulfill parameters EL1 and EL2 and EL5. Furthermore, in respect of meeting the condition of power-saving, SDN-based solutions fulfill parameters PL1 and PL5.

5.2.2. Access Level Contribution (Solutions)

By way of access network approaches, one of the significantly suggested notions, and also parallel to LTE twofold connectivity, is the idea of a phantom cell [85]. It permits the UE to camp its CP (control plan) on a macro cell (MC), whereas its DP is controlled at the small cells (SCs), which reside inside the coverage of the above-stated MC. Specifically, the MC handles the radio resource management for the phantom cells, and henceforth, the CN signaling is shunned for the duration of handovers between the phantom cells [55]. Next, due to the softwarization of the whole network, the information interchange among various OSI layers, i.e., implementation of the cross-layer scheme, is eased [102,103]. However, to grasp the cross-layer approaches, substantial modifications will be required to the protocol stack of the software architecture [84,103]. Another result of the softwarization procedure is cloud-RAN (C-RAN), which offers on-demand provision of network resources at the access level, subject to the user and network situation [104–106].

Furthermore, the proposed multi-RAT solutions are broader categorizations for the various RAT selection procedures (RSSI-centric, genetic algorithm-centric, optimization-based, fuzzy logic-centric, etc.) [27,88,90], since, from our previous discussions, it is noticeable that RSSI-based approaches are simple but not optimal for the selection of RAT due to non-consideration of other factors such as network or user policies, network burden, and backhaul conditions. Hence, another context-aware method is proposed [92,107], which will provide an optimal solution for even real-time situations.

Another critical aspect required to be mentioned here is that the above-stated handoff decision may be performed either by the combined effort of UE and the network (hybrid approach), by the network, or only by the UE. Along with the consideration of each aspect and in light of the discussion in Table 2 of Section 3, we describe the advantages and drawbacks of each approach (smart AP/RAT, cross-layer, radio access network as service, and phantom cell approach). These are as follows:

**Advantages of Smart AP/RAT choice**

-   Offers the capability to choose multiple RATs (AP) [87]
-   Use of the parameters such as the context of the user, the burden on the access point, the profile of the user, etc. [88–90]
-   Provides the option of selecting AP on the basis of per flow, per user, or per slice.
-   It is an enhanced (optimized) method of selecting the access point (AP) [5].

**Drawbacks of smart AP/RAT choice**

-   The QoS requirements in 5G are disturbed, owing to the complexity of the RAT selection algorithm [87].
-   The appropriate selection of the RAT requires rapid and accurate information about network conditions.

**Advantages of cross-layer approach**

- Allows for collaboration among multiple OSI layers, thus enabling multi-homing support in respect of efficient utilization.
- The distribution of network information among the different OSI layers is allowed [99].

**Drawbacks of cross-layer approach**

- Substantial software changes are required to the existing protocol structure [84].

**Advantages of radio access network as service (RANaas)**

- The handoff support is attained at the access level of the network [83].
- The physical connection is managed centrally, hence, the granularity service at channel-based and per-flow-based is attained.
- The network resources are assigned on an on-demand basis at the RAN level [80,81].
- On-demand handover execution is also possible [83].
- Supports permitting the devices to composite on more than one AP.

**Drawbacks of radio access network as service (RANaas)**

- Many modifications are required on the RAN side of the network [80,81].

**Advantages of the phantom cell approach**

- The access level network provides support at the time of handoff [60].
- Due to prevailing standards on MR-DC, it is easy to implement the operations [27].
- Multiple physical layer connections are possible.
- It offers the capability to permit per-user and per-flow granularity of service.

**Drawbacks of phantom cell approach**

- Signaling overload at CN at the time of inter-MC domain handoff.
- Smooth (no disruption) service availability is still an issue at the time of handoff among MC domains [27].

(A)  Considerations for smart AP/RAT choice

The smart RAT choice mechanism fulfills the dependability criterion based on parameter DL4 as mentioned in Table 2. Similarly, the adaptability parameters AL5 and AL4 can fulfill the criteria for adaptability. Next, the EL4 parameter can also fulfill the criteria of extensibility. Finally, PL1 and PL4 parameters of power-saving can be satisfied for power-saving criteria.

(B)  Considerations for cross-layer approach

The cross-layer mechanism fulfills parameters DL1 and DL4 for the dependability criteria. Next, for the adaptability condition, it fulfills parameters AL1 and AL4. Finally, the cross-layer mechanism fulfills parameter PL4 for the power-saving criteria.

(C)  Considerations for radio access network as service

The RAN-as-a-service model fulfills parameters DL4 for dependability criteria. Next, parameters AL1, AL2, AL4 and AL5 fulfill the adaptability criteria. Furthermore, parameters EL2, EL3, and EL5 fulfill the extensibility criteria. Lastly, parameters PL4 and PL5 are fulfilled for power-saving criteria.

(D)  Considerations for phantom cell approach

In the phantom cell approach, parameters DL4 and DL5 are fulfilled for the dependability criteria. Next, for the adaptability criteria, parameters AL2, AL4 and AL5 are satisfied. Further, regarding fulfillment of extensibility criteria, the phantom cell technique fulfills parameters EL2, EL3, EL4 and EL5. Furthermore, regarding power-saving conditions, the phantom cell technique fulfills the parameters PL1, PL2, and PL3.

### 5.2.3. Extreme Edge Network-Level Contribution (Solutions)

The extreme edge network-level solutions use the perspective of the D2D technique for smooth hand-off operations. Various studies [87,93] have proposed approaches to tackle the mobility of D2D pairs. Furthermore, for the exploration of D2D mobility, a Markov chain-based model [108] and a simulation-based model [93,109] have also been proposed. In the following discussion, we highlight the advantages, drawbacks, and considerations of the D2D approach concerning its applicability in light of the parameters mentioned in Table 2 of this article.

**Advantages of the D2D communication approach**

- Offers the option of decentralization at the time of MM.
- Offers D2D handoff controlling policies [92]

**Drawbacks of the D2D communication approach**

- The feasibility concerning delay and energy conservation for the MM mechanism has not been explored yet.
- We have the challenge of the increased number of control messages in the network.

**Considerations for the D2D communication approach**

Owing to the discussions and the advantages/drawbacks mentioned above, the D2D approach fulfills the parameter for dependability criteria.

Likewise, the parameter AL2 (offering mobility provision at the edge network level will execute in cooperation with core and access network-based approaches) fulfills the adaptability criteria. Next, the D2D approach fulfills parameter EL1 for fulfilling extensibility criteria. Furthermore, the D2D method does not satisfy any parameter from a power-saving perspective.

It has been derived from the literature and discussion that the D2D methods also contribute to MM through the CP support facility.

### 5.3. MM Mechanisms and 6G Network

We provide a short analysis in this section by observing Sections 5.1 and 5.2 along with the issues of state-of-the-art MM mechanisms for incorporating a 6G network. We also highlight prospective research areas for MM in 6G networks. In a 6G network, the NFV and SDN will offer the tools to provide the programmability aspect of the meta-surfaces, which will be rigorously tested in handoff situations [110]. The current networking paradigms allow for a time interval of 1–12 milliseconds for executing any programmability job (delay limitations, as stated in current 5G networks [8], for most of the services), but 6G networks will allow making more and more surfaces to be softwarized and orchestrated. In particular, as the number of surfaces/network nodes grows, more data must be administered to make suitable programmability decisions. These decisions must then be disseminated (orchestrated) to many network nodes (including meta-surfaces) to complete the task. As a result, the network programmability concern suffers from delay constraints.

Further, while meta-surfaces give the operator more freedom to program, that creates difficulty in handling SDN areas, NFV orchestration, and associated signaling. As a result, the current state-of-the-art SDN and NFV processes will fail in terms of effectiveness and compactness. It is also pointed out that the backhaul capacity will provide a substantial design problem because VLC technology can transfer data at a speed of more than 1 Terabyte/sec. The present backhaul systems are incapable of supporting large bandwidths [16]. Hence, it is vital to emphasize that the network will include mmWave, 4G-LTE Apps, VLC, and drone-based APs, which are the primary reasons for the additional complexity.

Furthermore, edge clouds offer low-latency access to cached material and planned resources; hence, deployment options will need to be reconsidered [111]. Reconsideration of the substantial limitations in existing state-of-the-art mechanisms for 6G networks and the essential publications in the field of 6G approaches [11,91,112,113] will open the following research areas.

1.  Selection of efficient RAT and AP, especially when both (UEs and APs) are mobile
2.  Selection of efficient RAT and AP, especially in a programmable situation
3.  Construction of a solid heterogeneous RAT approach, similar to the 4G/5G idea, given mmWave and Terahertz technology and their associated coverage issues
4.  Approaches for handling the massive increase in the number of messages during the handoff process, especially, at the time of handoff
5.  The consideration of dependability and handling of VLC links for MM mechanisms
6.  The impact of handoff on VLC, drone-based communication, and programmable scenarios
7.  The configurable setting permits the selection of optimum AP and RAT choices.
8.  The depiction of the computational complexity of optimizing methodology for user association.

It should be noted that the above-mentioned (Section 5.3) discussion is not complete in respect of 6G but is indicative of what portion of the MM mechanism in 6G networks needs to be explored for further research potential. Therefore, through the provided qualitative analysis, we uncover the hidden challenges (gaps) and solutions by researchers in the design aspects of 5G and future MM mechanisms, especially in the 6G network, in the following text (Section 5.4).

*5.4. Revealed Gaps, Prospective Solutions, and Expected Framework*

Our discussion and analysis from Sections 2–5 introduced the requirements from MM approaches, used as models for future MM systems, in order to fulfill the requirements mentioned in Tables 1 and 2. Next, we have investigated the legacy and current MM mechanisms for utilizing 5G and next-generation networks in Tables 3 and 4. Nevertheless, we have seen that the gaps in satisfying these requirements still exist. We have shown that none of the assessed methodologies fulfill the criteria of adaptability, extensibility, dependability, and power-saving measures completely (as a whole). Hence, in the following text, we highlight the gaps in the 6G network. These are as follows: (i) handovers and interference for 6G network, (ii) security, (iii) network slicing, (iv) massive numbers of handover signals, (v) consideration of context, (vi) architectural advancement costs, (vii) communication, control, localization, sensing, and power consumption, (viii) dynamic network topology, (ix) assurance of no (zero) latency [114], (x) reconfiguration of metasurface, (xi) 6G network and pertinent network protocol stacks, (xii) optimized handoff strategy [115,116], and (xiii) edge node pattern.

These identified gaps have been tackled by researchers in the following ways: (i) addition of control and data plane in D2D, (ii) utilization of clean state strategies, (iii) deep learning, (iv) provision of seamless service through edge computing, (v) effective signaling in the core network, (vi) incorporation of SDN and NFV for DMM, and (vii) MM on-demand bases [6].

To sum up the discussion (Section 5), we first provided the 5G architecture (based on services) and the classification of the different procedures that have been investigated, as seen in Figure 1 [117]. Succeeding that, we conducted qualitative research on 3GPP 5G MM approaches and initiatives regarding the worth of MM mechanisms for 5G and future networks. Subsequently, we presented Table 4 (by reviewing Table 2), in which we specified the method where each of the discovered mechanisms fulfill the criteria of dependability, adaptability, extensibility, and power-saving. We also included a list of the sources (reference articles) that helped to compile Table 4, as provided in Section 5. According to understanding from the above discussion and literature review as depicted in Table 4, we concluded that none of the discussed MM mechanisms completely fulfill the mentioned requirements (adaptability, dependability, extensibility, and power saving) of Table 1. We also revealed the various gaps/challenges of MM mechanisms for 5G and future networks. Further, we presented a brief discussion on possible solutions that can help describe these gaps/problems. Next, we demonstrated an innovative mapping between the gaps and potential solutions in Table 5. Moreover, we also mentioned the metrics for qualitative

investigation (requirements mentioned in Table 1). This, as a result, supported the completeness of our current discussion. Henceforth, in the subsequent section, by applying the implications from Sections 2–5 and Table 5, we suggest an architecture/framework for MM mechanisms.

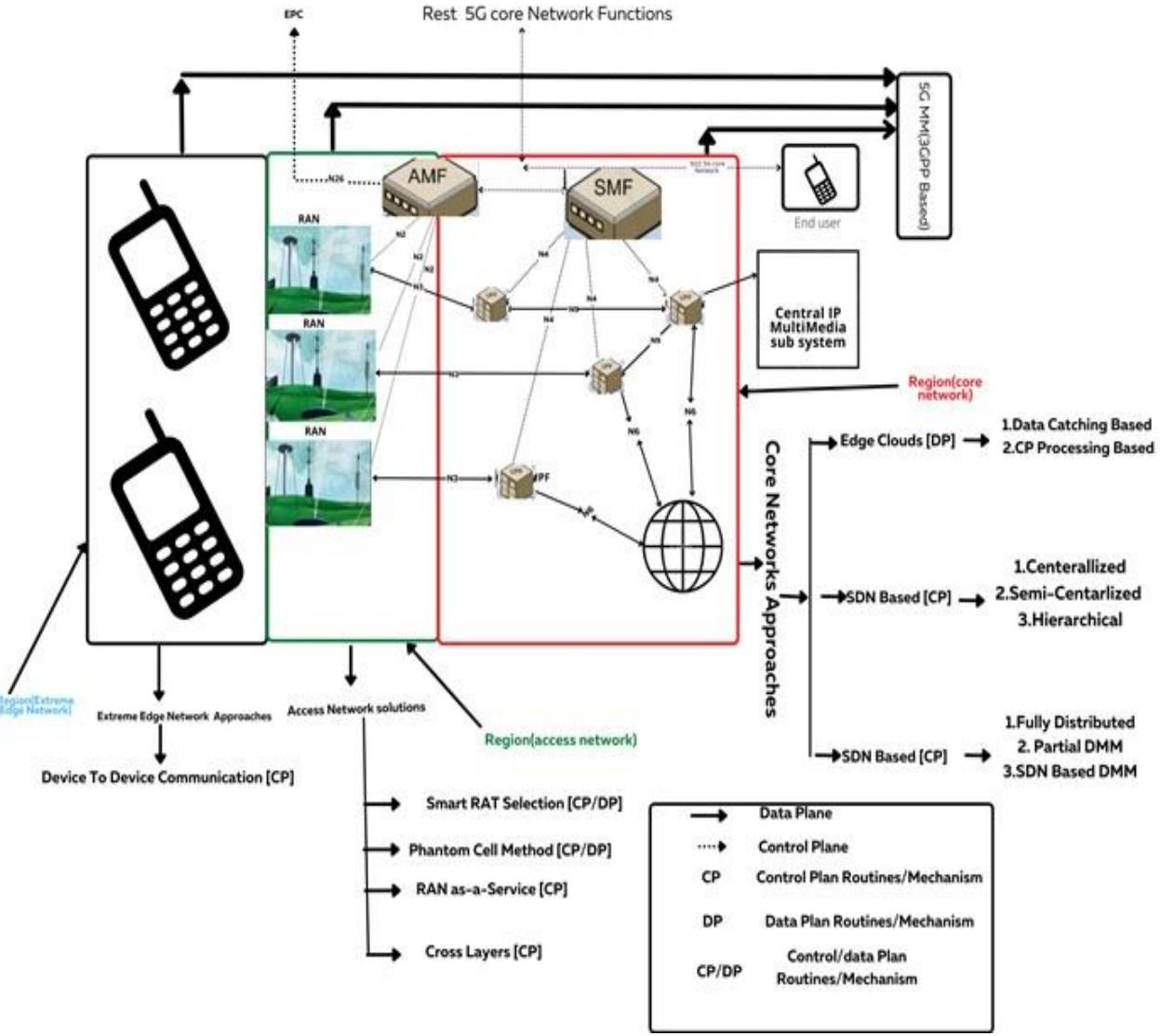

**Figure 1.** Organization of the innovative mobility management approaches in the 5G design.

Table 5. MM mapping of possible solutions to gaps/problems (mapping from Table 2).

| SNO | 1 | 2 | 3 | 4 | 5 | 6 | 7 | 8 | 9 | 10 | 11 | 12 | 13 |
|---|---|---|---|---|---|---|---|---|---|---|---|---|---|
| Gaps | HO signaling | Slice a network | Framework incorporation into MM solutions | Consideration of environment (context) | Architectural enhancement cost | Occurrence of very frequent handoffs | Security | Power saving | Handoff provision through Meta-Surface organization | Next Generation Handoff | Adoptive and dynamic Network Topology | Edge node alignment in next-gen network | Persistence of IP address |
| Possible Solutions | Smart CN signaling, and SDN-NFV mixed DMM | Mobility management on demand | Updating in Design Aspect | Mobility management on demand | Updating in design aspect | Deep learning | Effective signaling in the core network | Effective CN signaling and deep learning | Deep Learning | Service cont., CN sig by D2D and edge comp | Deep learning | Consideration of design aspect | Usage of clean state strategies |
| Remarks | Decreased HO signaling load by smart CN [4] and increased dependability by DMM. | Provision of tailor-made solutions to slice by on-demand strategy. | Need to consider the factors like efficiency and interruption at the time of handoff, while upgrading the design aspect | Suitable MM mechanisms are required for considering the context of the network, user, and application [118]. | Cost of infrastructure is considered | Deep learning can guess the parameters such as SINR, appropriate user association, etc., in a simple way without any complexity at the time of handoff. | It decreases the complexity and delays to ensure the security in the core network | Deep learning provides an ideal solution, and effective CN signaling provides power saving solution for mobility. | The demanded QoS can be provided by the support of deep learning with help of met surface arrangement at the time of handoff. | Seamless handoff is possible by using the stated solution. | The complex association can be incorporated efficiently with the help of deep learning at the time of handoff. | Need to consider all design gaps as well as parameters such as structure cost, benefits, etc. | The clean state method can support keeping a single IP address during whole communication with the destination server |
| Metrics fulfilled | DL2, DL1, EL5, EL5 | AL5, AL3 | EL4, PL1 | AL3 | EL4 | DL5, DL4, AL4, AL2, AL1, PL2 | DL4, EL2 | EL5, PL3 | DL5, DL4, AL2, PL2 | DL4, DL3, AL2, EL5, EL1 | DL4, DL5, AL1, AL2 | EL4 | DL3, DL4, PL4 |

### 6. Proposed 6G MM Framework

According to the discussion in the above sections and in the light of defined parameters (dependability, extensibility, adaptability and power consumption), we have presented our vision for 6G MM architecture in Figure 2. This architecture is characterized at the access, core and extreme edge network levels based on their impact on the network. The architecture mentioned in Figure 2 presents the strategies such as the RAN as service and multi-connectivity with the application of different access points and RATs efficiently at the access level. Next, it is also revealed that the RAT selection procedure may occur at the device or access level. This RAT selection process is also called the cell-less selection process, in which no overhead of handoff occurs from the adaptability, flexibility, and QoS point of view. Further, the core network techniques in Figure 2 are different from previous generations (G1 to G5) due to addition of the following features: (i) intelligent CN signaling, (ii) the mixture of distributed MM for NFV and SDN, (iii) network slicing pattern, (iv) 3D network paradigm, and (iv) low power consumption network operations (using energy harvesting circuit). Such features provide dependability, adaptability, extensibility and efficient use of battery power in a broader sense. Furthermore, the mentioned core network techniques/paradigms/methods must be accompanied by efficient CN (core network) signaling techniques.

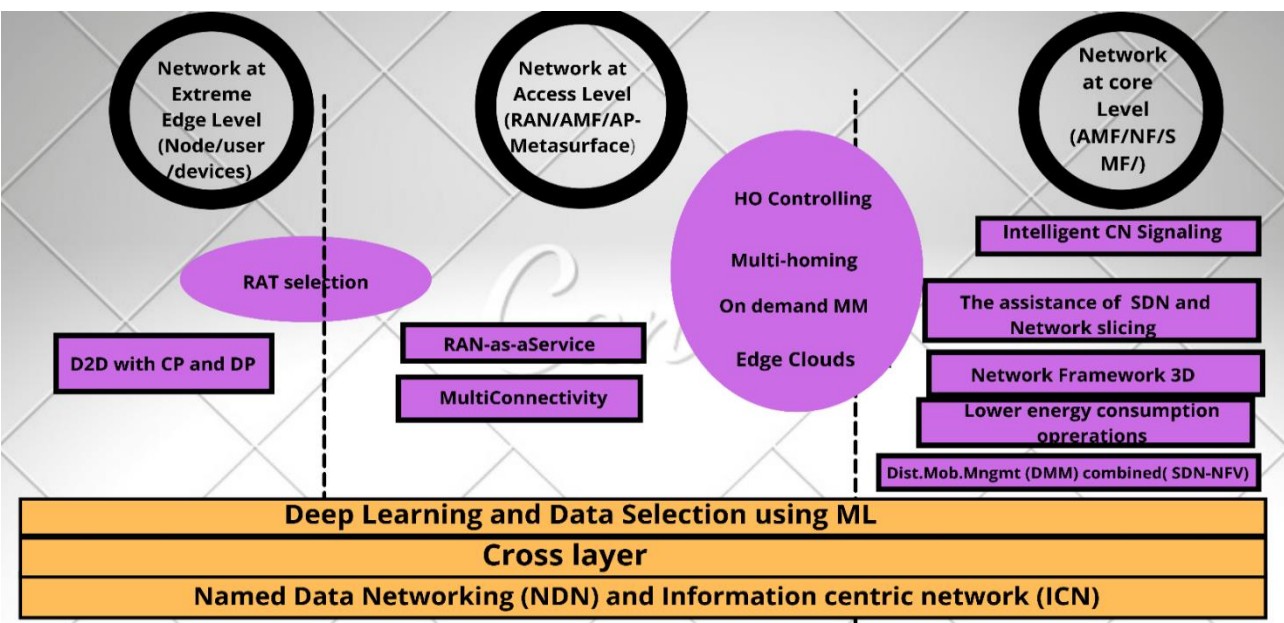

**Figure 2.** Future 6G MM structure.

Similarly, some other techniques or paradigms are used in the core or access network or collaboration of both (access level and core network level). Those techniques (core network or access network or both) can fulfill dependability and adaptability criteria. Likewise, D2D methods are expected to deliver additional support for mobility at the device level (edge network) with the help of CP and DP options. Along with these methods (core, access, and extreme edge network level), as described in Figure 2, the named data networking and information-centric support will also be offered at all levels; therefore, it helps to maintain IP addresses/prefixes during mobility while resolving destinations via names [119,120]. It has been observed that traditional IP address/prefix allocation techniques are not supposed to be altered by the addition of NDN-ICN ideas at all the levels of layers (core, access, and extreme edge); instead, they provide support at the top of this IP address allocation technique. Further, the deep learning and data selection using (machine learning (ML) techniques again provide support across the various levels by learning the complex characteristics of the user's mobility, the perspective of the network,

and an on-demand MM of the users. Furthermore, the cross-layer technique is overed across multiple levels. It applies the strategies by using the existing information at each level and supporting optimum MM decisions across the network.

To summarize the finding and discussion in the preceding sections, we employ the possible solutions for bridging the technology gap, as identified in Section 5.4, and specific strategies from state-of-the-art and legacy MM mechanisms, as specified in Sections 4 and 5. We have presented Tables 2–5. These Tables indicate the fulfillment of our proposed framework for the 6G network concerning the satisfaction of extensibility, adaptability, power consumption and dependability conditions. Finally, the suggested architecture in Figure 2 will also meet all of the requirements described in Table 1, hence offering a comprehensive solution. With this in mind, we summarize the key results of this work and conclude this paper as follows.

## 7. Conclusions

This article helps the research community to invent or re-invent the most appropriate MM mechanisms by following the capabilities concerning gaps/challenges and possible solutions provided for the complex scenario of 5G and 6G networks. In Section 2 and Table 1, we suggested the critical design criteria and functional requirements that need to be measured at the time of developing 5G and future MM solutions. Next, in Section 3 and Table 2, we highlighted the various parameters required to be fulfilled by future MM mechanisms. These parameters were adaptability, dependability, extensibility, and power consumption. Similarly, in Section 4, we concluded that the assessed legacy MM mechanisms are unable to fulfill the criteria of adaptability, dependability, extensibility, and power usage as a whole. However, the present standards, which were investigated in Section 5, are capable of offering improved performance (according to our defined criteria) to future MM solutions. From this investigation, we have inferred a comparison and presented the results in Tables 3 and 4 in the form of a qualitative analysis. From this qualitative analysis, a number of advantages and drawbacks of the legacy and current state-of-the-art mechanisms were identified, which can certainly be understood by the researchers. Next, we inferred from the findings that there is no existing MM mechanism that can be used entirely (as a whole) to fulfill the stated parameters of 5G and 6G MM mechanisms. Hence, it is evident from the finding that a generic MM mechanism for 5G and future networks may be undefinable. Therefore, Section 5.4 represents the gaps and challenges that are still required for planning, designing, developing, and deploying next-generation MM mechanisms. Additionally, in Section 6, we also briefly discussed the possible gaps and challenges that exist in MM techniques. Furthermore, the worth/efficiency of our current research was mentioned in Table 5, in which we drew a state-of-the-art mapping between possible strategies and challenges. Finally, we have proposed an innovative framework for the 6G MM mechanism based on our findings and defined parameters.

**Funding:** This work was funding supported by the Programme: SME Growth Romania–Priority ICT, Contract Greensoft No. 2020/548467, Project name: AI FERODATA.

**Conflicts of Interest:** The authors declare no conflict of interest.

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
