# Peer review of "The Challenges and Compatibility of Mobility Management Solutions for Future Networks"

_applsci, doi:10.3390/app122211605_

Round 1

Reviewer 1 Report

The paper presents MM mechanisms (6G Network) by following the capabilities concerning gaps/challenges and possible solutions provided for the complex scenario of 5G and 6G networks.

The paper performs an extensive state of the art review of the relevant 5G technologies and several 6G related mechanisms, but appears more as a bulk detail of information, rather than an extensive analysis on the related technologies. The figures are of very low resolution and need to be updated. Additionally, both figures appear to be directly copied from an external source and they need to be recreated in an alternate form. Many paragraphs are formatted wrong, which does not help the reader. Major syntax corrections and revision is required.

Given these points, the authors need to revise the scope of the paper and offer a vision of the 5G/6G landscape overall, in order for the manuscript to not just be so reference focused. All figures need to be recreated in an original form, according to the proposed style format. And the text needs to be revised both syntax and format-wise. Apart from these points I think the review is quite extensive and with a good direction it could form as an adequate SoTA paper.

Reviewer 2 Report

1. In Figure 1, please enter the reference from which you downloaded the figure.

2. Lacks background of your research

3. Related work must be rearranged

4. Lacks meaning of the abbreviation D2D

Round 2

Reviewer 2 Report

Your effort is evident. You have analyzed numerous relevant references in this area. You have systematized many problems/challenges nicely. It can be considered as your contribution. It would be good if, in further investigations, you propose a much more concrete solution and check its effectiveness.